# A Systematic Literature Review. Relationships between the Sharing Economy, Sustainability and Sustainable Development Goals

**Andrei Boar** [1,2,*]**, Ramon Bastida** [2] **and Frederic Marimon** [1] 

[1] Departament d'Economia i Organització d'Empreses, Facultat de Ciències Econòmiques i Socials, Universitat Internacional de Catalunya, 08017 Barcelona, Spain; fmarimon@uic.es

[2] Barcelona School of Management, Universitat Pompeu Fabra, 08008 Barcelona, Spain; ramon.bastida@bsm.upf.edu

[*] Correspondence: Andrei.boar@bsm.upf.edu; Tel.: +34-935-42-18-00

**Abstract:** The sharing economy is an umbrella term for different business models that seem to have a positive impact on sustainability. To achieve sustainable development, the UN has created 17 Sustainable Development Goals (SDGs) as an instrument and guide for countries and individuals. This paper sheds light on the relationships between the sharing economy, sustainability and SDGs through the content analysis of 74 papers from the database Web of Science that consider these factors through the topics of the environment, consumer values, business characteristics and urban impact regarding sectors such as mobility and accommodation. Results express that the sharing economy helps to achieve all SDGs. However, further research is needed because of the novelty of the topic and the existence of many gaps. The sharing economy has a positive effect on the dimensions of sustainability from the economic, social and environmental points of view.

**Keywords:** sharing economy; Sustainable Development Goals; sustainability; mobility; accommodation

## 1. Introduction and Theorical Framework

Sustainable development was defined for the first time in 1987 as "development that meets the needs of the present without compromising the ability of future generations to meet their own needs" [1]. Most of the standards address sustainability issues through economic, environmental and social dimensions [2]. How can we promote sustainable development?

The sharing economy is usually related to sustainability, and it is framed as: (a) an economic opportunity, (b) a more sustainable form of consumption and (c) a pathway to an equitable and sustainable economy [3]. There is no common definition for the sharing economy, but some authors have tried to shed light on this topic, which has been used as an umbrella term for a great variety of organisational models [4]. One definition is that of an ecosystem, whose intermediary companies utilise online platforms to facilitate and lower the cost of the for-profit transactions of giving temporary access—without the transfer of the ownership—to the idle resources of consumers in peer-to-peer networks that it has created, because of the trust built among its members who may be individuals or businesses [5].

The sharing economy can be placed on three foundational cores: (a) access economy, (b) platform economy and (c) community-based economy [6]. They define each concept as follows:

Access economy: initiatives sharing underutilised assets (material resources or skills) to optimise their use.

Platform economy: intermediation of decentralised exchanges among peers through digital platforms.

Community-based economy: coordination through non-contractual, non-hierarchical or non-monetised forms of interactions (work, exchange, etc.).

Other concepts are usually used as synonyms for the sharing economy, but they are not used properly. Some of them are characteristics of the sharing economy or different activities that are included in that "umbrella":

- Digital economy: Digital economy means economic activity, with the help of mobile technology and the internet of things (IoT), that results from billions of everyday online connections among people, businesses, devices, machines, data and processes [7].
- Peer economy: Peer-to-peer economy refers to the business between customer and customer without any intermediaries. They can buy and sell products and services from each other [8].
- Gig economy: Gig economy means temporary, project-based and flexible jobs. Companies that hire independent contractors and freelancers instead of full-time employees are part of this so-called gig economy [8].
- Collaborative consumption: This is the peer-to-peer based activity of obtaining, giving or sharing access to goods and services, coordinated through community-based online services [9].
- Digital sharing economy: A digital sharing economy is a resource allocation system, based on sharing practices, that is enabled by information and communication technology (ICT) and coordinated through participation of individuals and possibly commercial organisations (businesses) with the aim of providing temporary access to resources that involve either direct or indirect monetary value [7].

According to previous definitions, it can be stated that the sharing economy helps to use resources inside a community without the need to consume or buy each time. The opportunity to share assets offers the possibility of only using an asset when it is really needed and after that, sharing it with other people. This kind of consumption means that property is less important and that not everybody that is willing to consume needs to own every asset; therefore, production can be lower than that without sharing. If production is reduced, it will definitively have a positive impact on sustainability. In the last few years, the sharing economy has increased as a new business model that will change consumers' relationship to a materialistic lifestyle [10] and it is expected to grow around 25% per year [11].

Sustainable Development Goals (SDGs) are a guide to achieving sustainable development by 2030. They were created in 2015, as the next step of the Millennium Development Goals. SDGs are defined by the UN as "integrated and indivisible, global in nature and universally applicable, taking into account different national realities, capacities and levels of development and respecting national policies and priorities. Targets are defined as aspirational and global, with each Government setting its own national targets guided by the global level of ambition but taking into account national circumstances" [12].

There are 17 SDGs with different objectives such as the reduction of inequality (SDG 10) or the improvement of quality in education (SDG 4). These 17 SDGs have 169 targets and many indicators that appear in the 2030 Agenda as a guide for countries to improve their sustainability. According to the Stockholm Resilience Centre, the 169 targets of the different SDGs can also be divided into three dimensions: economic, social and environmental, as shown in Table 1.

**Table 1.** Sustainable Development Goal (SDG) classification by sustainability dimensions. Adapted from [13].

| Dimension | SDG |
|---|---|
| Economy | SDG 8, 9, 10, 12 |
| Society | SDG 1, 2, 3, 4, 5, 7, 11, 16 |
| Environment | SDG 6, 13, 14, 15 |

Sharing economy business models that create sustainable value can be classified from an environmental, social and economic perspective [14]. They present different items that can be used as a reference for each dimension, and they are also cross-checked with the 17 SDGs:

- Environmental dimension: increasing resource efficiency, responsible use of resources, no harmful environmental impacts or emissions and increasing environmental well-being.
- Social dimension: safeguarding health and safety, respecting laws and regulations, respecting employees and stakeholders' rights and ethical principles, no harmful impacts and increasing social well-being.
- Economic dimension: increasing cost-efficiency, increasing profits and business opportunities, operational stability and risk reduction, increasing attractiveness, increasing economic well-being.

The Sustainable Development Goals are the blueprint to achieve a better and more sustainable future for all. They address the global challenges we face, including those related to poverty, inequality, climate change, environmental degradation, peace and justice [12].

According to the explanations of the United Nations, SDGs are a guide to achieving a more sustainable world in future years. The SDG framework helps to integrate social, economic and environmental dimensions for prosperity in the long term. Being that the sharing economy is a potentially significant contributor to sustainable development growth, it can contribute to achieving the relevant goals [15].

Some authors defend the sharing economy as a form of economic activity and expect that it will complement traditional forms of business, generating positive economic, social and environmental effects [16]. However, some studies suggest that while the sharing economy may contribute to addressing sustainability issues, its economic, social and environmental effects remain poorly understood [4].

Using a systematic literature review, this paper sheds light on the literature on the sharing economy, sustainability and SDGs. The main objective of the research is to identify the relationships that exist in the literature between the sharing economy and sustainability, and the sharing economy and SDGs. After that, a comparison of topics and content through both categories will be done to identify common points, differences and gaps.

The conclusions of this research can be used by sharing economy companies to modify their strategies and to include (or not) sustainability and SDGs in their activities. If this happens, it could have a positive impact on society. For academics, this paper sums up the actual knowledge on the topic and opens future lines of research.

The paper is structured in four sections. After the introduction, the methodology is explained. The third section presents a descriptive analysis of the selected papers. The fourth section analyses the papers by topic and, finally, the fifth section compares the papers and makes conclusions and recommendations.

## 2. Materials and Methods

In this paper, qualitative research has been done through a systematic literature review to identify the knowledge that exists on the impact of sustainability in the sharing economy and the effects of SDGs.

A systematic literature review must "comprehensively identify, appraise and synthesise all relevant studies on a given topic" [17]. Furthermore, it can be defined as "a synthesis of published materials that provide examination of recent or current literature, that may include research finding". It may or may not include comprehensive searching and a quality assessment. The synthesis is typically narrative and the analyses may be chronological, conceptual and thematic [18].

Tranfield et al. proposed a methodology of systematic literature review that includes three main stages in the analysis [19]:

a.     Planning the review. It includes the identification of the topic and the preparation of the review.

b.　　Conducting a review. It includes the identification of research, selection of studies, study quality assessment, data extraction and data synthesis.

c.　　Reporting and dissemination. It includes the report, recommendation and evidence.

According to the Tranfield methodology, the steps and activities carried out in the research were (see also Table 2):

1.　　Planning the review. Identification of research objectives.
2.　　Conducting a review.

　　　a.　　Materials search; in this step, keywords were identified, and databases selected.
　　　b.　　Selection; this step defines the inclusion and exclusion criteria.
　　　c.　　Descriptive analysis; for a general view, the selected papers are described and their quality analysed.
　　　d.　　Content analysis; the selected papers are studied in depth.

3.　　Reporting and dissemination. It includes a comparison of the literature, a discussion and gaps found for academics and practitioners.

**Table 2.** Methodology used in the paper, adapted from [19].

| **Research Objectives** |
| --- |
| To identify the theoretical framework of the sharing economy |
| To identify the main lines of research between the sharing economy and sustainability |
| To identify the main lines of research between the sharing economy and SDGs |
| To compare the literature between the sharing economy and sustainability of SDGs |
| **Initial Inclusion Criteria** |
| Documents included in the Web of Science (all databases) |
| **Setting the Inclusion Criteria** |
| (1) 'sharing OR collaborative OR platform economy' AND sustainability—2013 to May 2020 |
| (2) 'sharing OR collaborative OR platform economy' AND 'sustainable development goals OR SDG'—From 2015 to May 2020 |
| (3) Relevant documents from the bibliography of selected papers |
| **Applying the Exclusion Criteria** |
| After the reading of title and abstracts, only papers that were focused on the sharing economy and sustainability or the sharing economy and SDGs were selected. |
| **Content Analysis** |
| In-depth analysis and classification of papers by topics and sectors of activity. |
| Comparison between results of the sharing economy and sustainability or SDGs |
| **Critical Discussion and Futures Lines or Research** |

*Paper Acquisition and Selection Phase*

The papers were selected using the Web of Science database between 2010 and May 2020, as the sharing economy and SDGs are a new topic of research, and SDGs only appeared in 2015. However, relevant papers did not appear until 2016. The keywords used were: "sharing, collaborative, platform economy", "sustainability", "sustainable development goals" and "SDG".

A total of 311 papers were initially found in the Web of Science database, as shown in Table 3.

Table 3. Summary of data base research.

| Keywords Used | ('Sharing or Collaborative or Platform Economy') and ('Sustainability'). | ('Sharing or Collaborative or Platform Economy') and ('Sustainable Development Goals or SDG'). |
|---|---|---|
| Searched by . . . | Topic | All words. |
| Date range | 2010 to May 2020 | 2010 to May 2020 |
| Number of papers | 158 | 153 |

In order to focus on the papers that were more closely related to the research objective, three selection criteria were used for the selected papers, as reported in Table 4.

Table 4. Criterion of selection of papers for the content analysis.

| Criterion | Sharing and Sustainability | Sharing and SDG |
|---|---|---|
| First criterion: focus on the abstracts and title | Abstracts focusing on the sharing economy and sustainability have been included. | Abstracts focusing on the sharing economy and SDGs have been included. |
| Second criterion: focus on the papers | Papers focusing on the sharing economy and sustainability have been included. | Papers focusing on the sharing economy and SDGs have been included. |
| Third criterion: cited references | Papers not included in the Web of Science but that appeared in the bibliography of selected papers | |

The first criterion helped to select only papers that dealt with the sharing economy and sustainability or SDGs, and after that, these papers were analysed in depth. Some of them were excluded after the reading of the paper, however, others were included because they appeared in the bibliography of some selected papers. Finally, 61 papers were chosen that studied the sharing economy and sustainability, and 13 that studied the sharing economy and SDGs.

## 3. Descriptive Analysis

The aim of the descriptive analysis was to give a preliminary result on the papers focusing on the sharing economy and sustainability and SDGs. For the descriptive analysis of the selected papers, three perspectives are defined:

### 3.1. Papers by Time

According to the distribution of papers over time, as shown in Figure 1, we can see that consideration of the topic has been increasing in the few last years, and it seems that 2020 (data up until May) is going to be a year with more papers published on the topic. Prior to 2016, it is hard to find papers that are focused on sustainability or SDGs and the sharing economy, so we can say that it is a new topic, and it is growing.

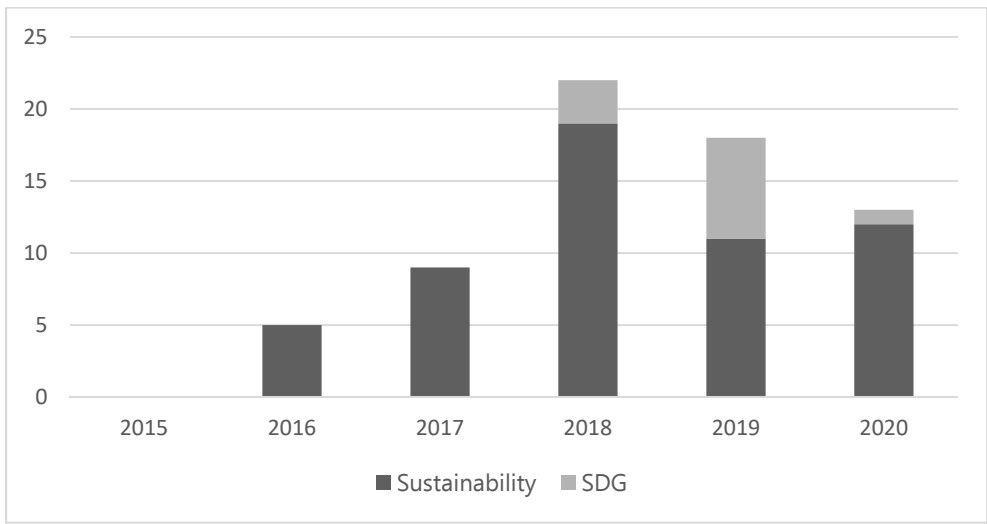

**Figure 1.** Distribution of papers over time.

### 3.2. Papers by Journal

The medium Journal Citation Reports (JCR) Impact Factor of the publications is 3.41, and they are distributed over 35 different journals. The journals with a greater number of publications on the topic were *Sustainability* (14 papers), *Journal of Cleaner Production* (7 papers), *Technological Forecasting and Social Change* (5 papers) and the *International Journal of Consumer Studies* (3 papers), as shown in Table 5. Papers published in *Sustainability*, as the journal with the most papers, analysed the impact of the sharing economy in the environment and in business.

**Table 5.** Distribution of papers by journals and impact factor.

| Journal | Papers Sustainability | | Papers SDG | | Impact Factor JCR—Index 5 Years |
|---|---|---|---|---|---|
| *African Journal of Hospitality, Tourism and Leisure* | | | 1 | 9.1% | 0.37 |
| *Agriculture and Human Values* | 1 | 1.8% | | | 3.41 |
| *Anthropocene Review* | | | 1 | 9.1% | 3.42 |
| *Brazilian Administration Review* | 1 | 1.8% | | | 0.4 |
| *Business Horizons* | 1 | 1.8% | | | 3.44 |
| *Competitiveness Review* | 1 | 1.8% | | | 2.47 |
| *Current Issues in Tourism* | 1 | 1.8% | | | 4.14 |
| *Ecological Economics* | 2 | 3.5% | | | 4.48 |
| *Economies* | 1 | 1.8% | 1 | 9.1% | 1.2 |
| *Energy Procedia* | 1 | 1.8% | | | 1.15 |
| *Environment and Behavior* | 1 | 1.8% | | | 4.26 |
| *European Transport Research Review* | 1 | 1.8% | | | 2.25 |
| *Food Policy* | 1 | 1.8% | | | 4.15 |
| *Interaction Design and Architecture(s)* | | | 1 | 9.1% | 0.64 |
| *International Journal of Consumer Studies* | 3 | 5.3% | | | 1.74 |
| *International Journal of Entrepreneurial Venturing* | | | 1 | 9.1% | 0.43 |
| *International Review of Retail, Distribution and Consumer Research* | 2 | 3.5% | | | 1.25 |
| *Journal of Business Research* | 1 | 1.8% | | | 5.35 |
| *Journal of Cleaner Production* | 6 | 10.5% | 1 | 9.1% | 7.1 |
| *Journal of Fashion Marketing and Management* | 1 | 1.8% | | | 1.97 |
| *Journal of Intellectual Capital* | | | 1 | 9.1% | 5.33 |

**Table 5.** *Cont.*

| Journal | Papers Sustainability | | Papers SDG | | Impact Factor JCR—Index 5 Years |
|---|---|---|---|---|---|
| *Journal of Marketing Theory and Practice* | 1 | 1.8% | | | 1.63 |
| *Journal of Sustainable Tourism* | 1 | 1.8% | 2 | 18.2% | 3.67 |
| *Local Environment* | 1 | 1.8% | | | 1.93 |
| *Management Science* | 1 | 1.8% | | | 4.53 |
| *Nature Communications* | | | 1 | 9.1% | 11.8 |
| *Psychology and Marketing* | 2 | 3.5% | | | 2.38 |
| *Resources Conservation and Recycling* | 1 | 1.8% | | | 8.08 |
| *Science of Total Environment* | 1 | 1.8% | | | 6.55 |
| *Sustainability* | 13 | 22.8% | 1 | 9.1% | 2.85 |
| *Sustainable Production and Consumption* | 1 | 1.8% | | | 3.77 |
| *Technological Forecasting and Social Change* | 5 | 8.8% | | | 4.85 |
| *Tourism and Hospitality Research* | 1 | 1.8% | | | 1.67 |
| *Transport Policy* | 1 | 1.8% | | | 3.77 |
| *Transportation Research Part D: Transport and Environment* | 2 | 3.5% | | | 4.75 |
| *Urban Policy and Research* | 1 | 1.8% | | | 1.81 |

### 3.3. Papers by Topic

The selected papers were classified in four different topics: environment, consumer value, business characteristics and urban impact. We can see that sustainability and the sharing economy can be found in all categories, the greatest being consumer value with 20 papers. However, as it can be seen in Figure 2, there are no papers about the impact of SDGs in the sharing economy on consumer value.

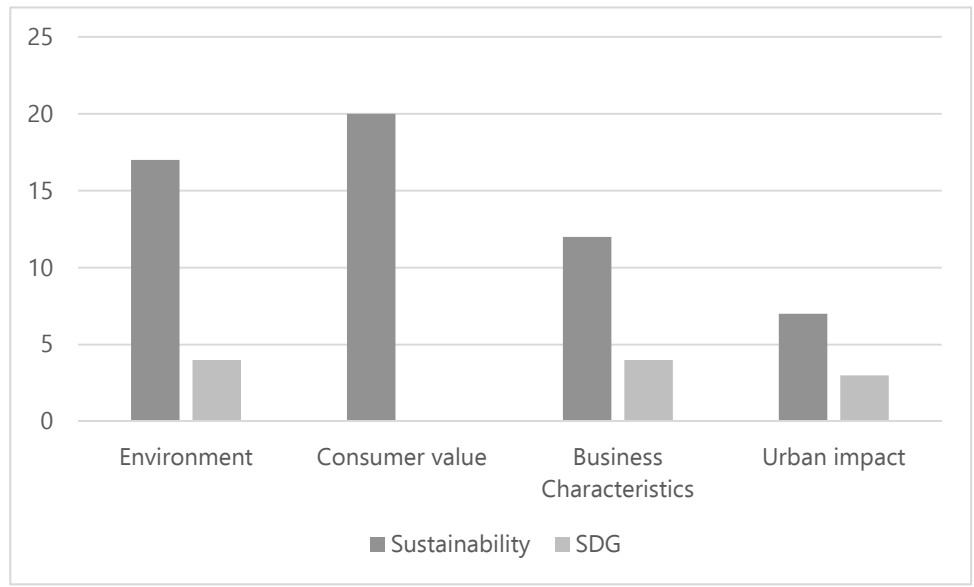

**Figure 2.** Distribution of papers by topic.

## 4. Content Analysis

Before entering into the explicit content of the papers, for a preliminary view of the topics, a bibliometric analysis was completed using the open-access software VOSviewer version 1.6.14 available at www.vosviewer.com with the papers that focused on the sharing economy and sustainability. VOSviewer allows the creation of relationships between the most relevant words in the literature and the identification of the main topics of research, classified by colours. From Figure 3, we can identify some topics of research in the literature:

- The circular economy and collaborative economy are two of the business models most studied in the sharing economy and sustainability.
- Sustainability is considered as one of the motivations of people for using the sharing economy, in addition to cost or trust.
- The major sector studied in the collaborative economy is accommodation through the company Airbnb.

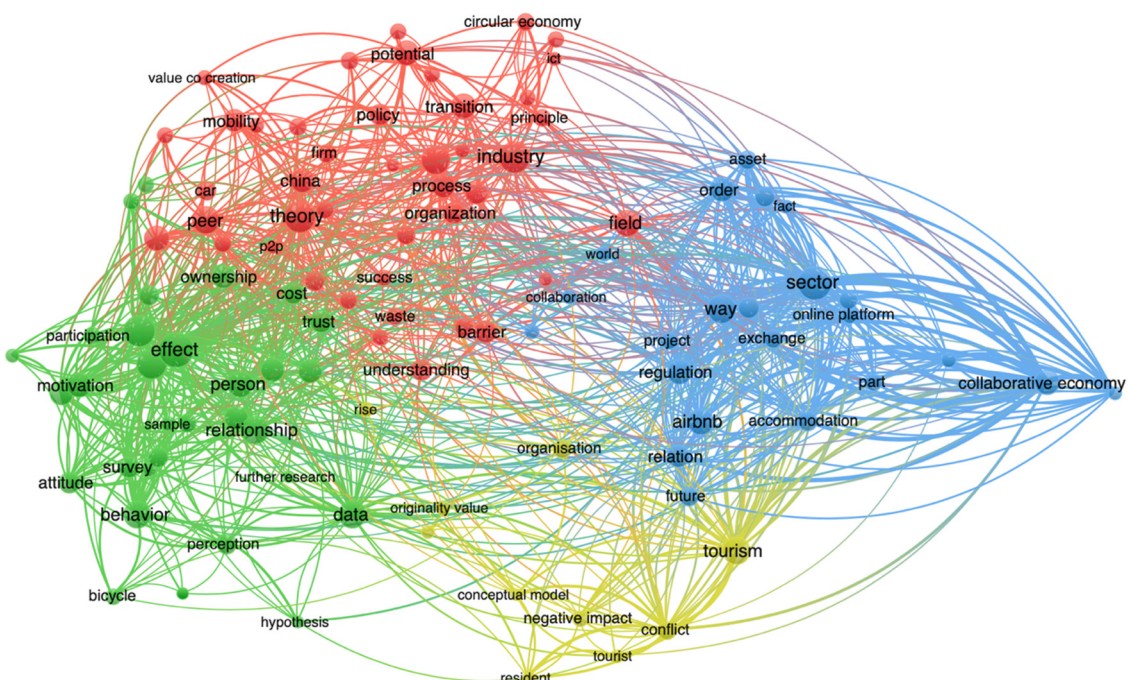

**Figure 3.** Bibliometric analysis of the sharing economy and sustainability performed with the software VOSviewer 1.6.14.

Literature tries to identify if Airbnb, a collaborative economy platform that offers accommodation for tourism, has a negative impact in the city, and if it creates a conflict between the tourist and the resident.

### 4.1. Impact of Sharing Economy in Sustainability

The sharing economy is an opportunity for sustainability. The possibility of using assets without the need of owning the property reduces the need for goods production and reduces waste. However, the impact of the sharing economy in the triple dimension of sustainability is not clear [4].

After reviewing 61 papers about the sharing economy and sustainability, four relevant topics were identified:

- Impact of the sharing economy in the environment.
- Value of sustainability for the decision of the customer about the use of the sharing economy.
- Business practices of sharing economy companies regarding sustainability. Urban impact of sharing economy companies.

The majority of the papers analyse the impact of the sharing economy in the environment using sectors such as mobility, bikes and clothes. In general, the sharing economy allows the use of under-utilised resources and, therefore, has an environmental benefit by reducing consumption [20]. Regarding shared mobility, it is stated that it reduces the negative impact on the environment and reduces polluting emissions and energy expenditure [21], being a transport element that should not be substituted by a particular car, but be complementary [22]. The reduction of a vehicle in the

family unit implies a 23% increase in the probability of shared car use in cities with high population density [23]. Shared bicycles, having the particularity of not emitting gases, have a positive impact on all environmental indicators. However, an exorbitant growth in its offer can have a negative impact due to the oversaturation of the service [24]. Use of shared clothing mainly implies a reduction of the waste generated by the consumption of first-hand clothing [25].

One of the main reasons cited by clients for using sharing economy platforms is sustainability, in addition to financial benefits, social experience or life quality [26–28]. Once again, the literature is focused on the effects of mobility and clothes sharing. Regarding car-sharing, key elements are knowledge, environmentalism, the possession–self link and involvement with cars [29]. In connection with the user of the platform the effects are different; for car owners, sustainability is a key factor in offering their car, however, for passengers it is irrelevant [30]. Regarding the second hand market, besides sustainability and economic benefits, another motivation for the use of the platform is distancing from the consumer system and the value of brands [31].

On top of this, companies are an essential element for achieving the three dimensions of sustainability. Criteria for analysing business practices in the company are: (1) using durable, quality goods; (2) intensifying use of goods; (3) enabling repair, take back and recycling of goods; (4) ensuring rental replaces purchase; (5) minimising transport and disposable packaging of goods; and (6) for transport, reducing the kilometres travelled by private vehicles [32]. One of the key elements of the social dimension is trust between users [33]. Moreover, companies need to know that while corporate social responsibility and investment recovery policies do affect the user's choice over whether or not to use a platform, internal policies do not have any impact at all [34]. With regards to entrepreneurs, the sharing economy can be an opportunity for them, and it is recommended that they apply an environmental corporate social responsibility that will affect the user's perception of sustainability and allow the creation of a brand [35].

Moreover, we can identify different models of the sharing economy in relation to mobility and food-sharing. Regarding sharing mobility, four models are defined as (1) peer-to-peer provision with a company as a broker, providing a platform where individuals can rent their cars when not in use; (2) short term rental of vehicles managed and owned by a provider; (3) companies that own no cars themselves but sign up ordinary car owners as drivers; and (4) on demand private cars, vans or buses and other vehicles, such as big taxis, shared by passengers going in the same direction [36]. What is more, models of food-sharing are (1) the "sharing for money" model, which is primarily a business-to-consumer for-profit model to reduce waste and, at the same time, generate revenue; (2) the "sharing for charity" model in which food is collected and given to non-profit organisations; (3) the "sharing for the community" model, which is a peer-to-peer model where food is shared amongst consumers [37].

The impact of the collaborative economy in the city has different effects. On the one hand, it causes gentrification in cities like Barcelona [38], but on the other hand, platforms such as Airbnb offer the possibility for growth of a new touristic model. The cultural heritage and location of the homes increases the number of users of the platforms [39]. In short, there is a discussion of what should prevail, either the conservation of space for locals or the promotion of collaborative economy platforms that are often used by tourists. What does seem clear is that the success of the collaborative economy will depend on whether or not it has support from the institutions [40]. The literature also asks that institutions be more agile in integrating different social agents in the collaborative economy to improve their efficiency, resilience and sustainability [41]. Institutions must create a regulation for the coexistence of both models, which avoids gentrification and harm to local inhabitants, but at the same time, allows platform users to make use of their services.

### 4.2. Impact of Sharing Economy in SDGs

Although SDGs are thought of as objectives for governments and states, companies are also one of the main agents responsible for the accomplishment of SDGs [42]. What is more, SDGs can also be

used as guidance for investments and opportunities in companies [43]. When the sharing economy was born, it was seen to have the potential to have a significant impact on SDGs [44] and to offer opportunities to companies. The sharing economy has promising outcomes for SDGs [45]. As an initial point of view, the sharing economy is expected to allow sustainable development according to SDGs [15]. The sharing economy is expected to be an instrument for sustainability, promoting economic growth and having a positive impact on society and the environment.

After reviewing 13 papers about the sharing economy and SDGs, four relevant topics were identified:

- Impact of the sharing economy in the environment.
- Business practices of sharing economy companies regarding SDGs.
- Urban impact of sharing economy companies regarding SDGs.
- Transversal category that includes different topics of all SDGs.

No papers were identified that analysed the consumer value of SDGs. Papers that analysed the relationships between the sharing economy and SDGs were focused in sectors such as accommodation and entrepreneurship. However, we can see that there were no papers focused on mobility.

The sharing economy has the potential to contribute to achieving all of the SDGs, relieving environmental pressures, promoting low-carbon emissions, reducing gender, education and income inequalities, stimulating sustainable consumption and production practices, using sustainable energy and transforming infrastructures and cities [46]. However, the sharing economy does not currently pay much explicit attention to environmental SDGs, such as clean water, clean energy, climate action, life below water or life on land [44].

Sharing enterprises should be encouraged to develop relationships with the local authorities and follow the related regulations in order to achieve long-term viability. Here, what is needed is more explicit acknowledgement by local and national governments of the importance of the sharing economy for achieving SDGs; the challenge is to better align the interests of both new and old businesses, local governments and the national economy. [47]. Another important factor that characterises the sharing economy is technology. Along these lines, the sharing economy can contribute to SDGs that describe digitalisation technologies, such as ICTs, as enablers of sustainable development [48]. On top of that, collaborative entrepreneurs can help to achieve SDGs and sustainable development in general [49]. Additionally, one model which needs to be studied further is collaborative consumption because this can offer more sustainable consumption options; understanding its application and impact is relevant to the SDGs [50]. Lastly, sustainable models that adapt sustainability and the collaborative economy should foster innovation to address social or environmental challenges and focus on at least one SDG [51].

Regarding urban impact, it is one of the most studied categories, which indicates that some sharing economy models, such as urban gardens, have the potential to achieve hunger reduction (SDG 2), to improve nutrition and sustainable agriculture practices (SDG 3) and to create sustainable cities (SDG 11). Urban gardens can also contribute to climate action (SDG 13) and to enriching local biodiversity (SDG 15) [46]. Positive aspects of the sharing economy in the accommodation sector were observed, including providing access to safe, affordable, accessible and sustainable transport systems for all (Target 11.2) and upgrading slums (Target 11.1). Negative effects were also noticed, particularly in clearly implementing Targets 11.6 (reducing the adverse per capita environmental impact of cities) and 11.7 (providing universal access to safe, inclusive and accessible, green and public spaces) [52]. Furthermore, the hospitality sector can make other contributions to SDGs 1 (no poverty), 5 (gender equality), 8 (decent work and economic growth), 9 (innovation), 11 (sustainable cities and communities), 12 (responsible consumption and production), 13 (climate action) and 16 (promoting peaceful and inclusive societies) [53].

The sharing economy can potentially contribute to four of the UN SDGs: sustainable economic growth (8); innovation (9); sustainable consumption and production (12); and peaceful and inclusive

societies (16) [44]. Trade-offs are inevitable within the SDGs; a focus on a certain form of industrial development, such as the collaborative economy, may generate employment, but its character may be different from other employment and may also have significant social and environmental trade-offs and rebound effects [44].

## 5. Discussion

The literature agrees that sustainability is one of the reasons that people use the sharing economy. Sustainability awareness is increasing in society and the sharing economy is a clear example. Other factors come into play, such as cost or quality of life, but it seems that the main reason is sustainability, which can impact the three dimensions of the SDGs. Regarding the environmental dimension, a reduction of negative impact on the planet can be found because of the reduction of emissions and waste; considering the economic dimension, the sharing economy has created new opportunities for companies but they require the intervention of authorities to create regulation in the sector; for the social dimension, the sharing economy improves quality of life but has a negative impact in the neighbourhoods of big cities such as Barcelona and Amsterdam because it creates conflict between tourists and local people, to the point that Amsterdam have banned touristic apartments in the city centre. However, when companies ask institutions for a new regulation, it is certainly not their idea. The collapse of some big cities is also an opportunity for the sharing economy, found through sharing mobility, which allows citizens to avoid the necessity of having one car per person, which is sometimes not an optimal solution in big cities.

The literature on the sharing economy and SDGs is still immature because of the novelty of the topic. However, authors agree that the sharing economy is an opportunity to work towards and to achieve all SDGs, because of the benefits of this business model. The sharing economy can help to achieve SDGs such as "economic growth" (8), "innovation" (9) or "sustainable consumption" (12). The effect on SDG 11, "sustainable cities", is ambiguous because it creates synergies and trade-offs with different targets within the same SDG. Urban gardens are also an important application of the sharing economy that can help to reduce the collapse of big cities, having a positive impact on SDGs such as "hunger reduction" (2) and "sustainable agriculture practices" (3).

It was found that the sharing economy has an impact on the three dimensions of sustainability also related to the SDGs. However, the papers did not focus on the environmental dimension; rather, they focused on the economic and social dimensions. Here we can find an important gap in the literature which needs to be complemented with the impacts on the environment according to the targets and indicators of environmental SDGs. Moreover, there were no papers that analysed the impact of the application of SDGs in a company on the final user. As we have seen, sustainability is one of the motivations of consumers for using the sharing economy, so an important research area is to find out the impact of each SDG in the final decision of the user. Furthermore, the literature is focused on accommodation and entrepreneurship, and some sectors that are important in the sharing economy, such as mobility or collaborative consumption, cannot be found when we talk about SDGs.

## 6. Conclusions

This paper proposed a systematic review of the sharing economy, sustainability and SDGs. The main objective of this research was to identify the relationships that exist in the literature between the sharing economy and sustainability, and the sharing economy and SDGs. After that, a comparison of topics and content through both categories has been done to identify common points, differences and gaps.

The descriptive analysis offers an overview of the papers included and identifies the topic as a new current of research that has been growing since 2016. Papers are published in many relevant journals, such as *Sustainability* and the *Journal of Cleaner Production*.

On the one hand, the main topics in the papers between the sharing economy and sustainability were the impact of the sharing economy on the environment, consumer value, business characteristics

and urban impact, and they were focused in sectors such as mobility and accommodation. On the other hand, the relation between the sharing economy and SDGs were focused on the environment and on business characteristics and practises, focusing on sectors such as accommodation and entrepreneurship.

For academics, more research is needed on this topic because the sharing economy allows sustainable development with few negative effects. Especially, the research needs to focus on the environmental dimension of sustainability, which is the less researched. Furthermore, we have little information about the effects of the sharing economy on SDGs and there are some gaps in the literature that need to be solved, such as the motivating influence of SDGs for the consumer, that will certainly aid companies in making decisions on their strategies. Moreover, sectors such as mobility are well studied in relation to sustainability, but the effects of the sharing economy on SDGs are not studied.

For practitioners, the sharing economy can be an excellent business model for achieving all SDGs and economic growth without negative effects on the planet and for trying to achieve an optimal sustainability that incorporates social, economic and environmental dimensions. Knowing the impact of SDGs on consumer value can be an important factor for their strategy in the next years. Especially, entrepreneurs will be one of the important actors in the business ecosystem in the future because they have the opportunity to innovate using technology (SDG 9) and can also directly promote sustainable business models.

The sharing economy can help to contribute to sustainable development according to SDGs, even if it is a new topic. More research will help academics and practitioners to improve the knowledge about the topic and will allow companies to adapt their strategies.

**Author Contributions:** The authors contributed equally to this work. All authors have read and agreed to the published version of the manuscript.

**Funding:** This article was written as part of a research project titled 'Improvement of quality in collaborative consumption companies: model, scale and loyalty (CC-Qual)' (REF: Tri 2018-096279-b-i00) funded by the Ministry of Science, Innovation and Universities of Spain within the aid programme for R&D "Retos de Investigación" project.

**Conflicts of Interest:** The author declares no conflict of interest.

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
