# Peer review of "A Systematic Literature Review. Relationships between the Sharing Economy, Sustainability and Sustainable Development Goals"

_sustainability, doi:10.3390/su12176744_

Round 1

Reviewer 1 Report

I think the manuscript is very interesting, and it meets the proposed aims. The title is appropriate and express in properly way the aim of the paper: “shed light on the relationships between the sharing economy, sustainability and SDGs.”.

The abstract fulfils its function of summarizing the paper’s content, and it remarks both the aims of the study and the key findings. By the way, its length is appropriate.

Regarding the introduction section, it is clear and well organized. In my opinion, the references are relevant, and they are correctly referenced. However, I feel that there is a small issue: personally, I find confusing the second paragraph of the introduction section. The use of numbers in the enumeration between brackets can lead to mistakes, because of the use of numbers in square bracket to cite the authors. I would suggest to the authors to use letters instead of number in the enumeration.

The description of the methodology is accurate, in my opinion. I would suggest to the authors to specify the version of the software used in the study, to ensure its replicability.

Both sections, results, and discussion, are correctly implemented and developed. The authors consider multiple angles to describe the results and their implications, and the conclusions are supported by them.

In general, the work seems interesting to me. The authors make a real contribution to the scientific literature, shedding light on the relationships between the sharing economy, sustainability, and SDGs, as they proposed.

I wish the authors good luck in revising their paper

Author Response

Dear reviewer, 

Many thanks for your comments. I have introduced your recommendations in the text. 

Regarding the introduction section, it is clear and well organized. In my opinion, the references are relevant, and they are correctly referenced. However, I feel that there is a small issue: personally, I find confusing the second paragraph of the introduction section. The use of numbers in the enumeration between brackets can lead to mistakes, because of the use of numbers in square bracket to cite the authors. I would suggest to the authors to use letters instead of number in the enumeration.

Thanks for that suggestion, I have changed numbers by letters and now it is much clear. I also make more changes in paragraph 3, also changing numbers by letters. 

The description of the methodology is accurate, in my opinion. I would suggest to the authors to specify the version of the software used in the study, to ensure its replicability.

I have specified the version of the software, which is VosViewer 1.6.14.

Thank you very much for your review. 

Reviewer 2 Report

The authors do not specify what type of research was used (eg qualitative research ...?) And the research methods are not clearly specified. Document analysis is a method of qualitative research. The acquisition of papers is not a methodological approach, but a concrete activity carried out according to a methodology. 

The objectives of the research are formulated in a stereotype and only partially meet the specific conditions of clarity, precision and measurability.

Consider it necessary to separate the Discussion of Conclusions into separate subchapters (Results and discussions in which the analysis part could be included; Conclusions)

Author Response

Dear Reviewer,

First of all, many thanks for your recommendations in the text. They have improved a lot the paper and now is much clear. I will answer to all recommendations point by point:

The authors do not specify what type of research was used (eg qualitative research ...?) And the research methods are not clearly specified. Document analysis is a method of qualitative research. The acquisition of papers is not a methodological approach, but a concrete activity carried out according to a methodology. 

Thanks for that comment. Now, we have included the qualitative research and also modify in introduction and methodology to include the research methods and steps as the acquisition of papers. You can find the changes in yellow. 

The objectives of the research are formulated in a stereotype and only partially meet the specific conditions of clarity, precision and measurability.

That is true and they were so generic. Thanks for your comment. Now, we have written them again with more precision and now is much understandable.

Consider it necessary to separate the Discussion of Conclusions into separate subchapters (Results and discussions in which the analysis part could be included; Conclusions)

Thanks for that recommendation. We have separe both parts and also separate conclusions between descriptive analysis and also include conclusions for academics and practitioners. 

Round 2

Reviewer 2 Report

116, 117 „This paper sheds light using a qualitative research on the literature on the sharing economy, sustainability and SDG” . Qualitative research is a type of research (along with quantitative research) and has specific methods. It is necessary to specify the method of qualitative research used by the authors (Record keeping, for example, or another method......Line 130).

The research methodology involves the choice of methods and, often, the justification of the choice in terms of how the method focuses on achieving the purpose and objectives of the research. The methodology cannot be presented through the research activities carried out. These activities or steps, especially in qualitative research, are chosen by researchers according to the research method they have chosen. So, according to the methodology, the authors carried out certain activities or steps (Lines 135-136).

Author Response

Dear reviewer, 

Thanks for your recommendation and explanation. It is important to be accurate in the description of the methodology and it was a weakness in our paper. Now, we have read some articles that have been very useful in this issue, particularly Tranfield et al. (2003), which is very popular and has been extensively quoted (more than 6,000 citations). We have included Tranfield methodology in the paper, that included the steps that we have followed for the systematic review. 

We attach the new version, in which the new paragraphs are added in yellow. We wait for additional feedback.

Our best wishes,

Round 3

Reviewer 2 Report

It's ok now.